# Agonistic CD40 Antibodies in Cancer Treatment

**DOI:** 10.3390/cancers13061302

**Published:** 2021-03-15

**Authors:** Dijana Djureinovic, Meina Wang, Harriet M. Kluger

**Affiliations:** Yale Cancer Center, Yale School of Medicine, New Haven, CT 06520, USA; dijana.djureinovic@yale.edu (D.D.); meina.wang@yale.edu (M.W.)

**Keywords:** CD40, agonistic antibodies, innate immunity, cancer

## Abstract

**Simple Summary:**

CD40 is a costimulatory molecule that is key for the activation of antigen-presenting cells and other innate immune cells. It plays an important role in anti-tumor immunity, and agonists of CD40 have been shown to eliminate tumors in both pre-clinical and clinical settings, alone and in combination with other treatment modalities. Here we assess the expression of CD40 and associations with other mediators of immunity in a variety of tumor types and review the potential of CD40 agonists for cancer treatment, given the promise of enhancing the interplay between innate and adaptive immunity.

**Abstract:**

CD40 is expressed on a variety of antigen-presenting cells. Stimulation of CD40 results in inflammation by upregulation of other costimulatory molecules, increased antigen presentation, maturation (licensing) of dendritic cells, and activation of CD8+ T cells. Here we analyzed gene expression data from The Cancer Genome Atlas in melanoma, renal cell carcinoma, and pancreatic adenocarcinoma and found correlations between CD40 and several genes involved in antigen presentation and T cell function, supporting further exploration of CD40 agonists to treat cancer. Agonist CD40 antibodies have induced anti-tumor effects in several tumor models and the effect has been more pronounced when used in combination with other treatments (immune checkpoint inhibition, chemotherapy, and colony-stimulating factor 1 receptor inhibition). The reduction in tumor growth and ability to reprogram the tumor microenvironment in preclinical models lays the foundation for clinical development of agonistic CD40 antibodies (APX005M, ChiLob7/4, ADC-1013, SEA-CD40, selicrelumab, and CDX-1140) that are currently being evaluated in early phase clinical trials. In this article, we focus on CD40 expression and immunity in cancer, agonistic human CD40 antibodies, and their pre-clinical and clinical development. With the broad pro-inflammatory effects of CD40 and its ligand on dendritic cells and macrophages, and downstream B and T cell activation, agonists of this pathway may enhance the anti-tumor activity of other systemic therapies.

## 1. Introduction

In the past decade, oncologic care has changed dramatically as immunotherapies have been developed for multiple tumor types. Immune checkpoint inhibitors targeting the programmed cell death protein 1(PD-1)/programmed cell death ligand 1 (PD-L1) pathway have significantly increased survival in randomized trials. Although durable responses are observed, most patients are inherently resistant or develop resistance to checkpoint inhibition over time [1,2]. Combinations of PD-1/PD-L1 axis inhibitors with inhibitors of cytotoxic T-lymphocyte-associated protein 4 (CTLA-4) have also been studied, and are now widely used for melanoma, renal cell carcinoma (RCC), and non-small cell lung cancer [3,4,5,6,7,8,9,10]. However, even with this dual approach, the majority of tumors are either resistant up-front or acquire resistance. Multiple mechanisms of resistance have been described, including impaired T cell function and lack of tumor-infiltrating lymphocytes in the tumor microenvironment (TME), which can be a result of diverse immune suppressive mechanisms [11,12,13]. Immense efforts are underway to circumvent immune suppression and restore T cell functionality and infiltration. Antigen-presenting cells (APCs), particularly dendritic cells (DCs) are crucial for the delivery of appropriate signaling that ultimately leads to T cell activation. A key mediator in this process is cluster of differentiation 40 (CD40), a costimulatory receptor molecule of the tumor necrosis factor (TNF) receptor superfamily. CD40 is predominantly expressed on APCs. Activation of CD40 by its ligand, CD40L (also called CD154), enables DCs to mature into professional APCs, often referred to as licensed DCs, and provides necessary signals for T cell activation as well as numerous other signals that induce immune activation [14,15]. Agonistic CD40 antibodies can mimic the binding of CD40L to CD40 and initiate downstream signaling that induces anti-tumor immunity. 

## 2. Expression of CD40 and CD40L and Downstream Signaling

CD40 is constitutively expressed as a transmembrane receptor on DCs, myeloid cells, and B cells, as well as several types of non-hematopoietic cells, such as endothelial cells, fibroblasts, epithelial cells, and certain types of malignant cells [16,17]. The main ligand for CD40, CD40L, is a transmembranous protein predominantly expressed on activated CD4+ T cells, activated B cells, and activated platelets, as well as monocytic cells, NK cells, mast cells, and basophils [15]. A soluble form of CD40L has been characterized, which is mainly derived from platelets. CD40L, in its soluble or membrane-bound form, is a trimeric molecule. Four receptors have been shown to bind to CD40L (αMβ2, αIIbβ3, a5β1, and CD40), but only binding to three of them (αMβ2, αIIbβ3, and CD40) induces downstream signaling [18]. The interaction between CD40L expressed on endothelial cells, and integrin receptor αMβ2 on rolling leukocytes promotes inflammatory recruitment of leukocytes resulting in atherosclerosis [19]. Soluble CD40L signaling through binding to αIIbβ3 on platelets is involved in the stability of arterial thrombi [20]. The binding of CD40 and CD40L plays a crucial role in both adaptive and innate immunity and engages several signaling pathways reviewed below that are involved in maturation, proliferation, and survival of DCs, production of inflammatory cytokines, and expression of costimulatory molecules. 

Engagement of CD40 by CD40L promotes the necessary clustering of CD40 receptors on the cell surface that initiates downstream signaling by one of two mechanisms: (a) activated CD40 binds Janus kinase 3 (JAK3) that in turn activates signal transducer and activator of transcription 5 (STAT5)-mediated transcription or (b) activated CD40 engages TNF receptor-associated factors (TRAFs) that can initiate nuclear factor-kappa B (NF-κB), mitogen-activated protein kinase (MAPK) and phosphoinositide 3-kinase (PI3K) pathways in immune cells. The specific downstream signaling resulting from CD40 activation is cell-type dependent [14,15,17,21].

JAK-STAT pathway signaling is involved in cytokine-mediated signal transduction. CD40-CD40L binding on monocytes recruits JAK3 that phosphorylates and activates STAT5 to initiate transcription of target genes [22]. JAK3-STAT5 signaling in monocytes induces transcription of tumor necrosis factor-alpha (TNF-α), interferon-gamma (IFN-γ), interleukin (IL) 1B, IL-6, IL-8, and IL-11 that influences the maturation of DCs. However, the interaction between CD40 and CD40L on B cells does not seem to activate this pathway [23,24].

Induction of NF-κB signaling pathway by binding of CD40 to CD40L induces activation of NF-κB inducing kinase (NIK). NIK activates IκB kinase α (IKKα), and IKKa in turn phosphorylates p100 to produce p52. P52 associates with relB and translocate to the nucleus to induce gene expression [25]. NF-κB signaling in B-cells impacts B-cell activation, differentiation, and proliferation [26]. In DCs, activation of the NF-κB pathway impacts upregulation of major histocompatibility complex (MHC) molecules and costimulatory molecules and maturation of DCs, but does not seem to have an impact on the survival of these cells [27]. 

MAPK signaling plays a role in the initiation of innate immunity in various cells to activate adaptive immunity ([28]. p38) MAPK is important for immunoglobulin (Ig) class switch in B cells [29]. In DCs, CD40-CD40L induced p38 MAPK signaling is crucial for DC maturation and DC-mediated CD8+ T cell responses [30]. On monocytes as well as DC, this pathway stimulates the production of IL-12 [30,31].

Survival of DC is mainly mediated by the PI3K pathway [30]. On monocytes, PI3K-Akt signaling increases IL-10 production, and on DC it stimulates the production of IL-12. [30,32]. 

The downstream signaling effects of CD40/CD4L are therefore diverse and differ between cell types.

## 3. Effects of CD40 Activation on Immune Cells

CD40 activation on DCs and monocytes upregulates the expression of other costimulatory molecules such as CD80 and CD86. Their effects on MHC molecules enhance antigen presentation, license DCs, and activate CD8+ T cells [33,34,35,36]. CD40 activation also results in increased expression of adhesion molecules such as intercellular adhesion molecule 1 (ICAM-1), several TNF receptor family ligands (e.g., 4-1BBL/CD137L, GITRL, and OX40L), and multiple additional receptor molecules (4-1BB/CD137, GITR, and OX40) that are involved in T cell activation [37]. Mature DCs and macrophages secrete IL-12, which is important for the cytotoxic activity of T cells, the polarization of the T helper type 1 (Th1) phenotype, and may lead to T cell-independent anti-tumor responses by activation of NK cells [33,38,39]. The interaction of CD40-CD40L on macrophages promotes anti-tumor effects by the production of TNF-α, nitric oxide, or by antibody-dependent cellular cytotoxicity (ADCC) [40,41]. CD40 activation on B cells increases their activation, antigen-presenting capacity, and proliferation by the formation of germinal centers, immunoglobulin isotype switching, somatic hypermutation, and the production of plasma and memory cells. Some studies have reported that CD40 activation on B cells can promote anti-tumor effects by providing support to already existing T cell immunity or by functioning as potent APCs and generating effector T cell activity. In addition, plasma cells secrete antibodies directed towards autoantigens overexpressed by tumor cells that may induce ADCC [42]. On certain malignant cells, CD40 activation on tumor cells themselves can trigger apoptosis and result in inhibition of tumor growth [43,44]. Thus, activation of CD40 can prevent tumor growth indirectly by anti-tumor immune cell responses or directly by tumoricidal activity such as apoptosis and ADCC (Figure 1). 

CD40/CD40L binding, therefore, activates multiple innate immune cells, including DCs, macrophages, NK cells, and B cells, resulting in CD8+ and CD4+ T cell stimulation. Activation of cytotoxic and effector T cells leads to cell kill and apoptosis. 

## 4. Expression of CD40 in Various Tumor Types

The important role that CD40 signaling plays in anti-cancer immunity prompted us to explore genes that may be included as potential co-targets with a CD40 agonist-based therapy. We used RNA sequencing data from The Cancer Genome Atlas (TCGA) database to analyze the correlation of CD40 expression and that of other protein-coding genes (*n* = 20,501) in samples from 534 clear cell renal cell carcinomas (ccRCC), 456 cutaneous melanomas, and 178 pancreatic adenocarcinomas, selecting tumor types that tend to be responsive to immunotherapy (melanoma and ccRCC) versus pancreatic adenocarcinoma which is resistant. Applying a cutoff Spearman correlation rho of ≥ 0.3 and an adjusted *p*-value of <0.001, we found that 198 genes in ccRCC, 361 genes in melanoma, and 935 genes in pancreatic adenocarcinoma were correlated with CD40 gene expression. The 50 genes with the highest correlation to CD40 in each cancer type are listed in Table 1. Eight genes were shared between all three cancer types (KIAA1949, IL-27RA, PSMB8, TRIM21, VAMP5, CASP4, TYMP, and STX4).

Interleukin 27 receptor subunit alpha (IL-27RA) is a receptor for the cytokine IL-27. Although the role of IL-27RA in cancer is unknown, preclinical studies have shown that IL-27 inhibits tumor cell growth [45,46]. Tripartite motif-containing protein 21 (TRIM21) is involved in the intracellular antibody-mediated proteolysis pathway. Depletion of TRIM21 promotes migration and invasion of tumor cells and induces changes in gene expression that regulate epithelial-mesenchymal transition. Higher expression of TRIM21 is associated with longer survival in breast cancer [47]. Several CD40 correlating genes that were shared between pancreatic adenocarcinoma and melanoma are involved in antigen presentation including TAP1, TAP2, HLA-DPB2, and HLA-DR85. We also found correlations between the expression of CD68, CD33, ICAM1, TGFB1, toll-like receptors TLR1, TLR2, TLR5, and CXCL10 and CD40. C-X-C motif chemokine ligand 10 (CXCL10) is secreted by several cell types in response to IFN-γ resulting in the attraction of T cells, monocytes/macrophages, NK cells, and DCs [48,49,50].

ccRCC and melanoma shared three genes that correlate to CD40 (OASL, GNLY, and IFI35). The number of genes that correlate with CD40 in both ccRCC and pancreatic adenocarcinoma was the highest (*n* = 82) and include several C-X-C chemokine receptors including CXCR3, which is the ligand for CXCL10, as well as CXCR4 and CXCR6. We found several genes that are involved in the cytotoxic activity of T cells and NK cells including granzymes GZMM, GZMA, GZMH, and natural killer cell granule protein 7 (NKG7), that were similarly correlated with CD40 expression. NKG7 is a cytolytic-related protein expressed in NK cells and T cells, preferentially those polarized to Th2 direction [51,52]. A recent study found that NKG7 and GNLY were overexpressed in patients that responded to anti-PD-1 and CTLA-4 in malignant melanoma [53]. Although the correlation between CD40 and co-expressed genes of interest needs to be validated at the protein level, this analysis raises many possibilities for future mechanistic studies to further understand the effects of CD40/CD40-L activation. Moreover, co-expressed genes, such as selected chemokines and their receptors or TLRs, might be good targets for co-activation with agonists of CD40 or CD40L.

## 5. Pre-Clinical Studies Supporting Development of Agonistic CD40 Antibodies for Cancer

Agonistic CD40 antibodies have been shown to effectively inhibit tumor growth and prolong survival in several tumor models. Although CD40 agonistic antibodies alone have had some effect, the advantage of CD40 agonism has been in combination with other treatments, such as chemotherapy, immune-based therapy (checkpoint inhibition, colony-stimulating factor 1 receptor (CSF-1R) inhibition, and TLR3 agonists), and anti-angiogenic antibodies. Some of the preclinical studies have provided the rationale for the ongoing clinical trials, of which the majority are in combination with other therapies, as discussed below.

### 5.1. CD40 Agonists in Combination with Chemotherapy and the Sequencing of Treatments

Several pre-clinical studies have demonstrated the anti-tumor activity of CD40 agonism and chemotherapy [54,55]. Besides apoptosis and necrosis, chemotherapeutic agents such as paclitaxel and doxorubicin can stimulate pro-inflammatory changes in the TME [56,57]. Chemotherapy can result in the release of intracellular antigens from damaged or dying cells that are taken up by APCs which activate CD8+ T cells [58]. Gemcitabine suppresses myeloid-derived suppressor cells (MDSCs), upregulates expression of immune accessory molecules and adhesion molecules (e.g., CD80, CD86, CD40, ICAM-1), and increases tumor-specific T cell responses in a mouse model of oral cancer [59]. In patients with mesothelioma, gemcitabine increases the number of NK cells and proliferating T cells but decreases regulatory T cells and MDSCs [60]. In murine studies of breast cancer, melanoma, and pancreatic cancer, paclitaxel leads to a shift of tumor-associated macrophages (TAMs) to an inflammatory phenotype [61]. Similarly, in patients with ovarian cancer paclitaxel activates inflammatory macrophages. [56] Taken together, these studies and others provide evidence for inflammation induced by chemotherapy, which might be harnessed for an anti-tumor inflammatory response that may be enhanced by CD40 agonists.

Pre-clinical studies of chemotherapy in combination with CD40 agonists have confirmed this approach. The synergy between gemcitabine and CD40 agonism was observed in murine mesothelioma [54]. The combined treatment resulted in an increase of both CD4+ T cells and CD8+ T cells but only CD8+ T cells were necessary for the observed anti-tumor activity. Several studies have been done on the genetically engineered KrasLSL-G12D/+, Trp53LSL-R172H/+, Pdx1-Cre (KPC) model, in which mice develop spontaneous pancreatic ductal adenocarcinoma [55]. CD40 agonism combined with gemcitabine/nab-paclitaxel reduced tumor growth more effectively than either modality alone. Regulatory T cells in the TME were markedly reduced after anti-CD40 treatment, independent of chemotherapy. However, the addition of chemotherapy increased T cell production of proinflammatory cytokines IFN-γ, TNF-α, IL-2, shifted the myeloid compartment toward the M1 (inflammatory) phenotype, and the T cell subsets in favor of Th1-type signature [55]. In a similar study, CD40 agonism and gemcitabine-induced macrophage-dependent anti-tumor immunity in KPC tumors [62].

The importance of timing and administration sequence with agonistic CD40 antibodies and chemotherapy has been demonstrated [54,63]. CD40 agonism administered before gemcitabine or concurrent with gemcitabine, but not after, resulted in lethal toxicity in murine mesothelioma and KPC mice as an effect of macrophage activation [54,63]. These studies of treatment sequence support clinical trials of CD40 agonists followed by chemotherapy, rather than concurrent therapy.

### 5.2. Synergism with Other Macrophage Modulating Drugs

CSF-1 signaling through its receptors, IL-34 or CSF-1R, stimulates recruitment, differentiation, and survival of pro-tumorigenic M2 macrophages [64]. It is believed that inhibition of CSF-1R signaling reduces the number of pro-tumorigenic M2 TAMs and reprograms remaining TAMs towards an inflammatory M1 phenotype, and therefore represents an attractive immunotherapeutic approach. CSF-1R inhibition combined with CD40 agonism is synergistic in various murine tumor models. In an autochthonous mouse melanoma model that is inherently resistant to checkpoint blockade and displays low baseline levels of CD8+ T cell infiltration, the combination of anti-CD40 and CSF-1R inhibition increased the expression of TNF-α in TAMs and IFN-γ production in T cells. Secretion of both TNF-α and IFN-γ was necessary for the best response in this model [65]. Similar anti-tumor activity was observed in a colorectal cancer model (MC38) treated with anti-CD40 and CSF-1R inhibition [66]. These studies underscore the synergizing effects of anti-CD40 and CSF-1R inhibition and their potential for disabling immunosuppressive TAMs.

### 5.3. Combinations of CD40 Agonists and PD-1/PD-L1 Inhibitors, with or without Chemotherapy, and Effects on T Cell Infiltration

Immune checkpoint molecules expressed on T cells are involved in the regulation of T cells and immune tolerance [67]. Checkpoint inhibitors targeting CTLA-4 or PD-1/PD-L1 can restore impaired T cell functionality, but to produce an effective anti-tumor response, there is a need for pre-existing immunity [68,69,70].

In the MC38 colorectal cancer model, anti-CD40 therapy alone had minor effects on tumor growth. Tumor-infiltrating monocytes and macrophages demonstrated increased expression of PD-L1 after treatment with anti-CD40, which is one of the tumor-induced mechanisms for immune suppression. Co-administration of anti-CD40 and anti-PD-1/anti-PD-L1 inhibitors bypassed the induced resistance and resulted in the survival of over 50% of mice compared to less than 5% when anti-CD40 or anti-PD1 was administered alone [71]. Upregulation of PD-L1 on myeloid populations was also observed in an orthotopic pancreatic cancer model following treatment with anti-CD40. The combination of anti-CD40 and anti-PD-L1 prolonged survival compared to either treatment alone [72]. The most effective combination in this study was CD40 agonism, checkpoint inhibition, and radiation. The observed anti-tumor effect was dependent on both T-cells and myeloid cells. CD40 agonism given before anti-PD-1 therapy increased the anti-tumor response by activating T cells in a breast cancer model that exhibited an anti-PD-1 resistant immune cell phenotype [73]. Immunological memory was established in the KPC model, but not in a melanoma model (B-16F10), when mice were re-challenged with tumor cell injection [74]. Adding checkpoint inhibition to anti-CD40 and chemotherapy increased the ratio of CD8+ T cells compared to T regulatory cells further, and was superior to anti-CD40 and chemotherapy. The combination of anti-PD-1, anti-CD40, and chemotherapy provided long-term complete remission and survival in tumor-bearing KPC mice, a process that was T cell-dependent [75]. The combination of anti-CD40 and anti-PD-1 was superior compared to either treatment alone in models of cholangiocarcinoma and the effect was dependent on DCs, macrophages, and T cells [62]. Taken together, these studies support further development of CD40 agonists in combination with immune checkpoint inhibitors.

### 5.4. Combinations of CD40 Agonists and Other Immune Modulatory Approaches

CD40 agonists have been used as adjuvants for cancer vaccine therapies. They have also been used in combination with TLR agonists. These approaches have been used in the preclinical setting as well as in clinical trials, and have promise in tumors amenable to local injection [76,77].

### 5.5. Combinations of CD40 Agonists and Inhibitors of Angiogenesis

Vascular endothelial growth factor (VEGF) stimulates tumor growth by a variety of mechanisms. VEGF secreted by tumor cells promotes immune tolerance by inhibiting DC maturation and preventing the presentation of tumor antigens to T cells [78,79]. VEGF has also been shown to increase the MDSCs population in vivo [79]. The inhibitory effect that VEGF has on DC maturation is achieved by inhibition of NF-κB pathway signaling in hemopoietic progenitor cells [79,80]. Angiopoietin-2 (Ang2) also functions as a pro-angiogenic factor, predominantly expressed by tumor endothelial cells [81]. Ang2 stimulates the production of IL-10 and the proliferation of T regulatory cells [82]. Drugs that target VEGF/VEGFR signaling and/or Ang2 decrease tumor vascularity and reduce the immunosuppressive effects mediated by these factors. Anti-angiogenic therapies have been combined with CD40 agonists and appear to be superior to either class of drugs alone [83]. In a recently reported preclinical study, CD40 agonism was combined with antibodies to VEGFA or Ang2. Both anti-VEGFA/Ang2 and anti-CD40 independently skewed the inflammatory macrophage populations towards the inflammatory M1 phenotype and increased DC activation in the TME. The anti-tumor effects were further amplified when the two therapies were combined [84]. These preclinical studies support clinical trials combining CD40/CD40L agonists and anti-antiangiogenic drugs.

## 6. Development of Human Agonistic CD40 Antibodies for Clinical Use

Two mechanisms have been proposed to describe how agonistic human anti-CD40 antibodies mimic the clustering of CD40 molecules that is necessary for downstream CD40-CD40L signaling. One mechanism is mediated by cross-linking of the IgG1 monoclonal antibody fragment crystallizable (Fc) region through interactions with Fc gamma receptors (FcγR) [85,86]. The second mechanism offers the essential conformation through a hinge region of the IgG2 and occurs independently of FcγR crosslinking [87]. There are six agonistic human CD40 antibodies currently in early phase clinical development. APX005M, ChiLob7/4, ADC-1013, and SEA-CD40 are of IgG1 isotype and depend on crosslinking with FcγR for their agonistic activity, whereas selicrelumab and CDX-1140 are IgG2 and activate the CD40 molecule, independent of crosslinking. Besides the need for cross-linking, another important factor for the functionality of agonistic anti-CD40 monoclonal antibodies is epitope specificity [88,89,90,91,92,93]. In a recent study by Yu and colleagues, the relationship between the location of the epitope and isotype and the consequence for the biological activity of human anti-CD40 antibodies was investigated. Among the nine human anti-CD40 antibodies that they tested, three were derived from antibodies used in clinical trials; Lob714-m1, SGN40-m1, and CP870-893-m1 that were derived from ChiLob 7/4, SGN-40, and selicrelumab, respectively. Lob7/4 and SGN-40 showed similar agonistic activities that were largely dependent on FcγRII crosslinking. CP870-893-m1 displayed much stronger activity and was independent of FCγRII crosslinking. When the binding epitopes were mapped, ChiLob7/4, SGN-40, and CP870893 all bound similar regions within CD40. However, the regions that were critical for the binding were different. Without extensive crosslinking, only those antibodies that bind to cysteine-rich domain 1 (CRD1) were strong agonists. These results support the idea that monoclonal antibodies that bind epitopes closer to the cell membrane had weaker agonistic activity, irrespective of whether the agonism was achieved through the h2 isotype or FcγRII engagement. Yu and colleagues also reported that monoclonal antibodies that bound to CRD1 (e.g., ChiLob 7/4, SGN40, and CP870893) did not compete with the CD40L binding site whereas those antibodies that bound epitopes in CRD2 may interfere with CD40L binding [94].

### 6.1. Selicrelumab

Selicrelumab (also known as CP-870,893, RG-7876, RO7009789) is a fully human IgG2 antibody with potent agonistic activity and low binding affinity to human FcγRs. Selicrelumab does not bind the same site as natural CD40L and hence does not block the natural CD40L-CD40 interaction. To date, selicrelumab is the most extensively evaluated agonistic CD40 antibody in clinical studies [37]. Besides being tested as monotherapy, it has been evaluated in combination with chemotherapy, radiotherapy, immune checkpoint inhibitors, and anti-angiogenic therapy.

The first in human phase 1 study included 29 participants with advanced solid tumors that received single doses ranging from 0.01 to 0.3 mg/kg intravenously (i.v.) [95]. The maximum tolerated dose (MTD) was established as 0.2 mg/kg. The most common adverse events were transient grade 1–2 cytokine release syndrome (CRS), decrease in peripheral lymphocytes, monocytes, and platelets, and elevated serum liver transaminases and total bilirubin. Twenty-four percent of patients had stable disease. Partial responses were observed in 14% of participants overall and 27% of patients with melanoma. One of the participants received 9 subsequent doses (approximately every 8 weeks) and remained in complete response for more than 9 years [96]. A subsequent phase 1 study evaluated selicrelumab administered every week in 27 patients with advanced solid tumors. In this study, no objective responses were observed and the best clinical response was stable disease (SD) in 26% of patients. Although fairly well-tolerated, weekly selicrelumab as monotherapy had little clinical activity in patients with advanced cancers in other studies [97]. Several early-phase trials have evaluated selicrelumab in combination with chemotherapy [98,99]. The therapeutic activity was observed when selicrelumab was administered together with gemcitabine in chemotherapy naïve patients with surgically unresectable pancreatic ductal adenocarcinoma (PDAC) [98]. The partial response (PR) rate was 19%, SD was seen in 52%, and progressive disease (PD) in 20% of patients [98,100]. Similar response rates were observed with carboplatin and paclitaxel in solid tumors [99] Selicrelumab together with cisplatin/pemetrexed had activity in patients with malignant pleural mesothelioma resulting in a PR rate of 40% and SD rate of 53% [101]. The combination of selicrelumab and an anti-CTLA-4 blocking antibody (tremelimumab) was evaluated in patients with metastatic melanoma [102]. The overall response rate (ORR) was 27.2% with 10% CR and 18% PR. The most common adverse event was grade 1-2 CRS [102]. Although no randomized data are available, this response rate is higher than expected for anti-CTLA-4 alone. The first in-human study evaluating anti-CD40 and CSF-1R inhibition in solid cancers reported SD as the best overall confirmed response in 40.5% of patients. [103] An early-phase study is currently evaluating selicrelumab in combination with a-PD-L1 (atezolizumab) in solid tumors (NCT02304393). Selicrelumab with anti-Ang2-VEGF bispecific antibody (vanucizumab) or anti-VEGF (bevacizumab) has been evaluated in solid tumors (NCT02665416) (Table 2). Early phase studies evaluating selicrelumab and atezolizumab with chemotherapy are underway in PDAC (NCT03193190) as well as selicrelumab and atezolizumab with bevacizumab in triple-negative breast cancer (NCT03424005) and colorectal cancer (NCT03555149).

### 6.2. CDX-1140

CDX-1140 is a fully human IgG2 monoclonal antibody that binds outside the CD40L binding site, independent of FcR cross-linking, and is expected to synergize with the naturally expressed CD40L [107]. A phase 1 trial evaluating CDX-1140 with or without anti-PD-1 (pembrolizumab) or FMS-related tyrosine kinase 3 ligand (Flt3L) (CDX-301) in advanced cancers is ongoing (NCT03329950). Interim data from the combination of CDX-1140 and CDX-301 have shown upregulation of cytokine responses and suggests that adding CDX-301 to CDX-1140 may enhance the activity. [108] Early phase studies are underway to evaluate CDX-1140 together with radiotherapy and CDX-301 in breast cancer (NCT04616248), lung cancer (NCT04491084), CDX-1140 with or without CDX-301 in pancreatic cancer (NCT04536007), and CDX-1140 together with TLR3 agonists as an adjuvant for a mutated neoantigen peptide vaccine in patients with melanoma (NCT04364230) (Table 2).

### 6.3. APX005M

APX005M is a humanized rabbit IgG1 monoclonal CD40 antibody in which the Fc portion has a mutation that is designed to facilitate interaction with Fc receptors to mediate antibody cross-linking. APX005M binds with high affinity to human CD40 and blocks the binding of natural CD40L. Currently, eleven early phase clinical studies are evaluating APX005M alone or in combination with immune checkpoint inhibitors or chemotherapy. Two phase 2 studies are evaluating APX005M in combination with chemotherapy in advanced soft tissue sarcoma (NCT03719430) and resectable esophageal and gastroesophageal carcinoma (NCT03165994). Manageable safety profiles and promising antitumor activity were reported for patients with metastatic pancreatic adenocarcinoma (PR 58%) evaluating APX005M in combination with chemotherapy [104]. A phase 2 study is investigating APX005M in combination with anti-PD-1, one in NSCLC and metastatic melanoma (NCT03123783), and a second trial is evaluating local injection of APX005M in combination with pembrolizumab in unresectable melanoma (NCT02706353). Interim data from the metastatic melanoma cohort of NCT03123783 demonstrated a good safety profile and promising anti-tumor activity [109]. A phase 1/1b study evaluating APX005M in combination with cabiralizumab (CSF-1R inhibitor) with or without nivolumab in NSCLC, melanoma, and RCC patients that previously have failed to respond to anti-PD-1/PD-L1 therapy (NCT03502330) is ongoing. APX005M is being tested as monotherapy in pediatric CNS tumors (NCT03389802). Two phase 2 studies are evaluating the activity of APX005M administered at two different schedules in patients with unresectable melanoma (NCT04337931) and an ongoing trial is evaluating neoadjuvant therapy with or without CD40 agonistic APX005M for locally advanced rectal carcinoma (NCT04130854). A phase I study of APX005M with ipilimumab and nivolumab in treatment naïve patients with advanced melanoma or RCC is underway (NCT04495257) (Table 2).

### 6.4. ADC-1013

ADC-1013 (also known as JNJ-64457107 and mitazalimab) is a fully human IgG1 monoclonal antibody that binds to CD40 with high affinity dependent on FcR binding and crosslinking. ADC-1013 was designed to improve the binding to CD40 to enable intratumoral (i.t.) administration at a lower dose to reduce adverse events secondary to CD40 agonism [110]. The i.v. route has been the main modality of delivery of agonistic CD40 antibodies in clinical trials. Based on evidence from preclinical studies, i.t. administration might have the potential to improve antitumor activity and reduce adverse events, although systemic absorption of CD40 agonists still occurs [111,112]. Recently, the first in-human study of ADC-1013 evaluated i.t. and i.v. administration in patients with advanced tumors. I.t. administration into superficial lesions was well tolerated at clinically relevant doses [105]. An ongoing study aims to determine the recommended phase 2 dose and schedule of ADC-1013 administered i.v. to participants with advanced-stage solid tumors with a focus on NSCLC, pancreatic cancer, and cutaneous melanoma (NCT02829099). The drug is well tolerated and a PR was observed in one RCC patient, while ten patients had SD for at least six months [113] (Table 2).

### 6.5. SEA-CD40

SGN-40 (dacetuzumab) is a humanized IgG1 antibody that has been tested as monotherapy or together with rituximab and chemotherapy in hematological malignancies [114,115]. SEA-CD40 is derived from SGN-40 and the two antibodies have the same amino acid sequence, but the Fc region of SEA-CD40 is non-fucosylated to enhance binding to FcγRIIIa and improve agonist activity. A dose-escalation study is underway evaluating SEA-CD40 in patients with relapsed/refractory metastatic solid tumors which includes cohorts of combination therapy dose escalation with pembrolizumab with or without chemotherapy (NCT02376699). Clinical and biological activity has been reported with SEA-CD40 monotherapy, the drug is generally well tolerated [116] (Table 2).

### 6.6. ChiLob7/4

ChiLob is a chimeric IgG1 CD40 agonistic antibody that requires cross-linking for stimulation of CD40. A phase 1 study in patients with solid tumors refractory to conventional therapies established the MTD and the major dose-limiting toxicity (DLT) was a reversible liver enzyme elevation. ChiLob7/4 activates B cells and NK cells and can be administered safely i.v. (Table 2) [106].

Studies of the various CD40 agonistic antibodies remain in the early phase, and randomized data are not available yet. The activity has been seen as monotherapy, or in combination with other drugs, but optimal combinations and disease settings have yet to be established.

## 7. Summary

Pharmacological activation of CD40 by antibodies activates signaling cascades in a variety of immune cells, particularly dendritic cells, B cells, and macrophages, resulting in increased inflammation and activation of cytotoxic T cells. Co-stimulation of innate and adaptive immunity might result in enhanced anti-tumor activity, and CD40 agonistic antibodies have therefore been the subject of intense preclinical and clinical research. Pre-clinical models assessing CD40 agonists alone or in combination with radiation, chemotherapy, immune-therapy, or angiogenic therapy have largely demonstrated superior activity to either modality alone. Additional pre-clinical studies are needed to further determine the mechanisms of action of these drug combinations and refine the dosing and sequencing of drugs and regimens. Early phase clinical trials of CD40 agonists in a variety of tumor settings have demonstrated activity, and additional clinical studies are needed to determine whether CD40 agonists can be added to the expanding repertoire of cancer drugs to either increase response (and preferably cure rates) in the frontline setting or to treat patients whose tumors have progressed on other frontline regimens.

## 8. Conclusions

The importance of signaling of CD40/CD40L has been demonstrated in a variety of tumor models, and activation of CD40 affects multiple innate immune cells, providing the promise of harnessing CD40 as a therapeutic target in cancer. Early phase clinical trials have demonstrated that CD40 agonistic antibodies are generally safe, when administered alone and in combination with other anti-neoplastic modalities. Further studies are warranted to extensively evaluate CD40 agonists in cancer patients.

## Figures and Tables

**Figure 1 cancers-13-01302-f001:**
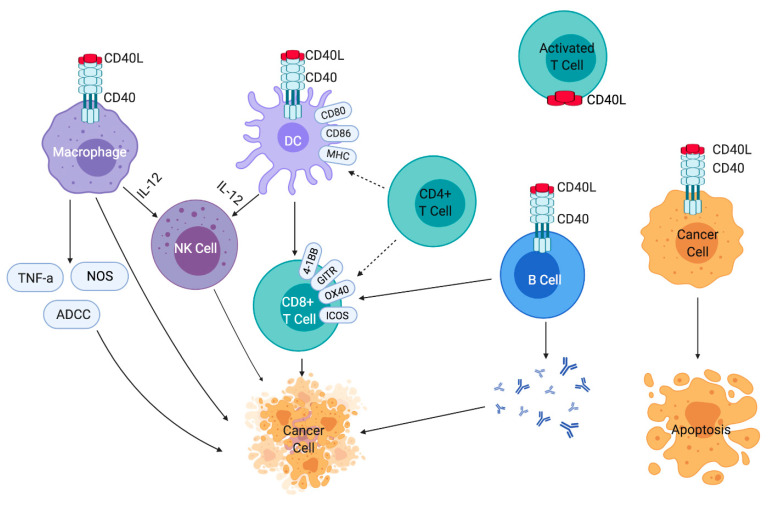
Effects of CD40-CD40L signaling on immune cells and cancer cells. CD40L is expressed on activated T cells. CD40-CD40L binding on macrophages results in the production of tumor necrosis factor-alpha (TNF-α), nitric oxide synthase, and antibody-dependent cellular cytotoxicity (ADCC). CD40-CD40L binding on dendritic cells and macrophages results in the secretion of interleukin 12 (IL-12), which is important for T cell and NK cell-mediated antitumor effects. CD40-CD40L activation on B cells supports T cell immunity and antibody production. Cancer cells may also be directly affected and undergo apoptosis after CD40 activation.

**Table 1 cancers-13-01302-t001:** The 50 genes with the highest correlation to CD40 in each cancer type.

Pancreatic Adeno Carcinoma	Spearman Rho	Adjusted *p*-Value ^1^	Melanoma	Spearman Rho	Adjusted *p*-Value ^2^	Clear Cell Renal Cell Carcinoma	Spearman Rho	Adjusted *p*-Value ^3^
ARHGDIB	0.57	2.85 × 10^−13^	COTL1	0.37	4.25 × 10^−15^	NSFL1C	0.35	1.76 × 10^−14^
TNFAIP8L2	0.56	4.79 × 10^−13^	LILRB5	0.37	4.25 × 10^−15^	WAS	0.35	1.76 × 10^−14^
WAS	0.56	4.79 × 10^−13^	BTN3A1	0.37	25.0 × 10^−15^	ZNF691	0.35	1.76 × 10^−14^
NCF1	0.56	4.79 × 10^−13^	C11orf93	0.37	4.25 × 10^−15^	TRADD	0.35	1.76 × 10^−14^
DOK3	0.56	6.83 × 10^−13^	SPAG4	0.37	4.25 × 10^−15^	EMP3	0.35	1.76 × 10^−14^
GPSM3	0.55	1.24 × 10^−12^	STX4	0.37	4.25 × 10^−15^	IL18BP	0.35	1.76 × 10^−14^
TREML1	0.55	1.24 × 10^−12^	IDO2	0.37	4.25 × 10^−15^	SQRDL	0.34	1.76 × 10^−14^
HLA-DPB1	0.55	1.27 × 10^−12^	EID3	0.37	4.25 × 10^−15^	CXCR4	0.34	1.76 × 10^−14^
FGD3	0.55	1.27 × 10^−12^	MOBKL1B	0.37	4.25 × 10^−15^	LOC84856	0.34	1.76 × 10^−14^
SIPA1	0.55	1.27 × 10^−12^	SYNE1	0.37	4.25 × 10^−15^	RPL11	0.34	3.17 × 10^−14^
ARHGAP9	0.55	1.35 × 10^−12^	JAK1	0.37	8.39 × 10^−15^	SH3BP1	0.34	3.17 × 10^−14^
LSP1	0.55	1.35 × 10^−12^	IL4	0.37	8.39 × 10^−15^	TRNAU1AP	0.34	3.17 × 10^−14^
SP110	0.55	1.46 × 10^−12^	MAN1A1	0.37	8.39 × 10^−15^	RSPH9	0.34	3.17 × 10^−14^
BTN2A2	0.55	1.57 × 10^−12^	CASS4	0.37	8.39 × 10^−15^	IRF1	0.34	3.17 × 10^−14^
CD53	0.55	1.76 × 10^−12^	GPR55	0.37	8.39 × 10^−15^	CD48	0.34	3.17 × 10^−14^
GMFG	0.55	1.76 × 10^−12^	HLA-DPB2	0.37	8.39 × 10^−15^	PRR14	0.34	3.17 × 10^−14^
CD72	0.55	1.93 × 10^−12^	C19orf36	0.37	8.39 × 10^−15^	TRIM55	0.34	3.17 × 10^−14^
SASH3	0.55	1.93 × 10^−12^	GRAP2	0.37	8.39 × 10^−15^	CCDC102A	0.34	4.65 × 10^−14^
MCOLN2	0.54	2.86 × 10^−12^	XBP1	0.37	8.39 × 10^−15^	EML3	0.34	4.65 × 10^−14^
CORO1A	0.54	2.86 × 10^−12^	BIN1	0.37	8.39 × 10^−15^	C5orf56	0.34	4.65 × 10^−14^
SIRPG	0.54	2.95 × 10^−12^	RAB8B	0.37	8.39 × 10^−15^	C21orf63	0.34	5.85 × 10^−14^
HLA-DMB	0.54	2.95 × 10^−12^	ATF7IP2	0.37	8.39 × 10^−15^	PRKD2	0.34	5.85 × 10^−14^
NECAP2	0.54	3.27 × 10^−12^	VAMP5	0.37	8.39 × 10^−15^	CYB5R3	0.34	5.85 × 10^−14^
BTK	0.54	3.27 × 10^−12^	GNAI2	0.37	8.39 × 10^−15^	ZNRD1	0.34	5.85 × 10^−14^
JAK3	0.54	3.30 × 10^−12^	RAB8A	0.37	1.25 × 10^−14^	RGS12	0.34	8.56 × 10^−14^
PARVG	0.54	3.55 × 10^−12^	KLHL33	0.37	1.25 × 10^−14^	PSMB8	0.34	1.11 × 10^−13^
KMO	0.54	3.69 × 10^−12^	C1orf38	0.37	1.25 × 10^−14^	RNF113A	0.33	1.33 × 10^−13^
ARHGAP25	0.54	4.35 × 10^−12^	MMP25	0.37	1.25 × 10^−14^	HYI	0.33	1.33 × 10^−13^
RAC2	0.54	4.35 × 10^−12^	CDRT4	0.37	1.66 × 10^−14^	RIBC1	0.33	1.33 × 10^−13^
PLEK	0.54	4.86 × 10^−12^	CYB5R4	0.37	1.66 × 10^−14^	BATF2	0.33	1.56 × 10^−13^
NAPSB	0.54	5.74 × 10^−12^	GBGT1	0.37	1.66 × 10^−14^	AP4M1	0.33	1.56 × 10^−13^
ITGAL	0.54	6.00 × 10^−12^	CAPZB	0.37	1.66 × 10^−14^	NDUFA4L2	0.33	1.80 × 10^−13^
FGD2	0.53	6.26 × 10^−12^	VAMP8	0.37	1.66 × 10^−14^	HSPB11	0.33	1.80 × 10^−13^
MYO1G	0.53	6.83 × 10^−12^	GNLY	0.36	1.66 × 10^−14^	RHOG	0.33	2.16 × 10^−13^
HMHA1	0.53	6.83 × 10^−12^	UGT2B15	0.36	2.08 × 10^−14^	DEF6	0.33	2.24 × 10^−13^
GNGT2	0.53	6.83 × 10^−12^	GPR84	0.36	2.48 × 10^−14^	TMEM44	0.33	2.24 × 10^−13^
PDCD1	0.53	6.83 × 10^−12^	TC2N	0.36	2.48 × 10^−14^	PPIE	0.33	2.24 × 10^−13^
CD52	0.53	7.21 × 10^−12^	CSF1	0.36	2.48 × 10^−14^	TMSB10	0.33	2.35 × 10^−13^
CTLA4	0.53	7.27 × 10^−12^	EMR4P	0.36	2.48 × 10^−14^	CLIC1	0.33	2.64 × 10^−13^
CTSW	0.53	7.27 × 10^−12^	TMEM155	0.36	2.48 × 10^−14^	INSL3	0.33	2.64 × 10^−13^
ARHGAP30	0.53	7.27 × 10^−12^	TMEM37	0.36	2.48 × 10^−14^	TICAM2	0.33	2.73 × 10^−13^
ANKRD58	0.53	7.27 × 10^−12^	SNX29	0.36	2.89 × 10^−14^	TBX21	0.33	2.73 × 10^−13^
STK17A	0.53	7.72 × 10^−12^	C12orf77	0.36	3.29 × 10^−14^	NFKB2	0.33	2.83 × 10^−13^
VAV1	0.53	7.72 × 10^−12^	NCRNA00204B	0.36	3.29 × 10^−14^	ALKBH2	0.33	3.24 × 10^−13^
CD2	0.53	8.50 × 10^−12^	KCNN3	0.36	3.29 × 10^−14^	KIAA1949	0.33	3.45 × 10^−13^
RASAL3	0.53	8.73 × 10^−12^	CLEC4M	0.36	3.70 × 10^−14^	IRF9	0.33	4.05 × 10^−13^

Adjusted *p*-value for the analysis in 1: pancreatic adenocarcinoma, 2: melanoma and 3: clear cell renal cell carcinoma.

**Table 2 cancers-13-01302-t002:** Clinical studies of CD40 agonistic antibodies in cancer treatment, alone or in combination with other treatments.

Study/ Reference	Phase	Status	CD40 Agonist	Cancer Type	Response
[95]	1	Completed	Selicrelumab	Solid tumors	14% PR, 24% SD
[97]	1	Completed	Selicrelumab	Solid tumors	26% SD
[98,100]	1	Completed	Selicrelumab + gemcitabine	PDAC	19% PR, 52% SD
[99]	1	Completed	Selicrelumab + paclitaxel and carboplatin	Solid tumors	20% PR,40% SD
[101]	1b	Completed	Selicrelumab + cisplatin pemetrexed	Mesothelioma	40% PR, 53% SD
[102]	1	Completed	Selicrelumab + tremelimumab	Melanoma	27% ORR, 9%CR, 18% PR
[103]	1b	Completed	Selicrelumab + emactuzumab	Solid tumors	40.5% SD
NCT03892525	1	Suspended	Selicrelumab + atezolizumab	B Cell Lymphoma	NR
NCT02304393	1	Completed	Selicrelumab + atezolizumab	Solid tumors	NR
NCT02665416	1	Completed	Selicrelumab + vanucizumab or bevacizumab	Metastatic solid tumors	NR
NCT03193190	1b/2	Recruiting	Selicrelumab + chemotherapy + atezolizumab	PDAC	NR
NCT03424005	1b/2	Recruiting	Selicrelumab + atezolizumab + bevacizumab	Breast Cancer	NR
NCT03555149	1b/2	Recruiting	Selicrelumab + bevacizumab + atezolizumab	CRC	NR
NCT04364230	1/2	Recruiting	peptide vaccine with CDX-1140 and TLR3 agonists	Melanoma	NR
NCT03329950	1	Recruiting	CDX-1140, CDX-301,pembrolizumab, chemotherapy	Advanced Malignancies	NR
NCT04536077	2	Not yet recruiting	CDX-1140, CDX-301	Pancreatic Cancer	NR
NCT04616248	1	Not yet recruiting	CDX-1140, CDX-301, radiotherapy, and Poly-ICLC	Breast Cancer	NR
NCT04491084	1/2	Not yet recruiting	CDX-1140, CDX-301 and SBRT	NSCLC	NR
NCT04520711	1/1b	Not yet recruiting	TCR-transduced T cells, CDX-1140, pembrolizumab	Epithelial Neoplasms	NR
NCT02482168	1	Completed	APX005M	Solid tumors	NR
[104]	2	Completed	APX005M, gemcitabine, nab-paclitaxel, nivolumab	Pancreatic adeno ca.	58% PR, 30% SD
NCT03123783	2	Ongoing	APX005M + nivolumab	Melanoma, lung cancer	NR
NCT03502330	1	Ongoing	APX005M + cabiralizumab w/w.o nivolumab	RCC, melanoma, NSCLC	NR
NCT03719430	2	Recruiting	APX005M + doxorubicin	Advanced soft tissue ca.	NR
NCT03165994	2	Recruiting	APX005M + radiation, paclitaxel, carboplatin	Esophageal cancers	NR
NCT02706353	2	Recruiting	APX005M + pembrolizumab	Metastatic melanoma	20% PR, 20% SD
NCT03389802	1	Recruiting	APX005M	Pediatric CNS tumors	NR
NCT04337931	2	Recruiting	APX005M	Melanoma	NR
NCT04130854	2	Recruiting	Neoadjuvant therapy with and without APX005M	Rectal adeno ca	NR
NCT04495257	1	Recruiting	APX005M, nivolumab and ipilimumab	Melanoma and RCC	NR
[105]	1	Completed	ADC-1013 (i.t vs. i.v)	Solid tumors	i.t well tolerated
NCT02379741	1	Completed	ADC-1013	Solid Tumors	NR
NCT02829099	1	Ongoing	ADC-1013	Solid tumors	1% PR, 11–14% SD
NCT02376699	1	Recruiting	SEA-CD40, pembrolizumab, nab-paclitaxel	Advanced malignancies	NR
[106]	1	Completed	ChiLob 7/4	Solid tumors	SD
NCT01561911	1	Completed	ChiLob 7/4	Advanced malignancies	NR

Toxicities are covered in the body of the article. CR: complete response, CRC: colorectal cancer, i.t.: intratumoral, i.v.: intravenously, NR: not reported, NSCLC: non-small cell lung cancer, PDAC: pancreatic ductal adenocarcinoma, PR: partial response, SD: stable disease.

## Data Availability

The RNA sequencing data are from the Firehouse legacy dataset for skin cutaneous melanoma, kidney renal clear cell carcinoma, and pancreatic adenocarcinoma generated by the TCGA Research Network: https://www.cancer.gov/tcga.

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
