# Peer review of "Agonistic CD40 Antibodies in Cancer Treatment"

_cancers, 2021, doi:10.3390/cancers13061302_

Round 1
Reviewer 1 Report
This manuscript is a concise and excellent review on CD40 agonistic antibodies for the treatment of cancer. Among new immunotherapies for the treatment of cancer, CD40 agonistic antibodies represent a powerful tool for combination therapies. The topic is relevant and timely.
A schematic figure showing the most common combinations and the cellular and molecular mechanisms implicated would be also very helpful.
The conclusions are consistent with the evidence and arguments presented, and this manuscript provides appropriate basic, pre-clinical and clinical information which is relevant for researchers in the field.
Minor point
The future of cancer treatment comprises the combination of therapies targeting different pathways. Thus I recommend to expand the section 5 on the rationale of each combination therapy. I would appreciate more detail into the mechanisms of action of these drugs, to improve the rational for each particular combination.
Reviewer 2 Report
The current review discusses the available activators of CD40 with regard to their anticancer activity. It focuses on the agonistic CD40 antibodies (APX005M, ChiLob7/4, ADC-1013, SEA-CD40, Selicrelumab and CDX-1140) which are evaluated in early phase clinical trials. The manuscript contains lots us useful information but needs significant editing to clarify the central message, carefully introduce the key concepts and underlying mechanisms and enhance the paper’s structure and general appeal.
- Please revise the last sentences of the summary and abstract for added clarity and more conclusive message. Currently, the inform the reader what is the review about, but do not provide any specific insights. This reduces the appeal of the paper.
- The overall paper would benefit from additional editing to highlight the central message, specific questions asked and conclusions. In its current form, the manuscript reads as a loose compilation of data from multiple original articles but its structure, especially the initial chapters lack a compelling structure. I also suggest eliminating unneeded repetitions (for example CD40 is introduced twice in parts 1 and 2 of the paper.
- The discussion of the clinical trials involving each of the discussed agonists is clear and compelling. In contrast, that background sections seems incomplete and insufficient to allow the reader understand how the system works. These paths deserve a thorough editing and additional narrative. I also suggest to revise Figure 1, which does not identify the natural sources of CD40L and ignores the involvement of Th cells in antitumor immunity.
- Please use consistent (and “symmetric”) nomenclature when introducing the key players. CD40 is is introduced as TNFRSF5, but no analogous description of CD40L is provided (as TNFSF5), which is just referred to as CD40L/CD154.Similar, in lines 74-75, the authors describe several other TNF receptor family ligands (CD137, GITR ligand and OX40 ligand and their receptors but do not specify their identity with regard to TNFSF/TNFRSF numbers. As a result, the selected nomenclature appears anecdotal.
- The paper needs a thorough read through to eliminate typos and incomplete words. For example: “Table 1. The 50 genes with highest correlation ion to CD40 in each cancer type” (underline is mine). Please spell our non-standard abbreviations when first used. For example “ccRCC”.
- CD40L is discussed only as a ligand of CD40. I miss additional discussion of the papers indicating that that it is also a signaling molecule.
- Selection of some of the references seem anecdotal. For example, the author cite several lesser (and later) papers of Bob Vonderheide on the activation of intratumoral macrophages and antitumor effects of CD40 agonists, but do not cite the original paper.
- The concluding paragraph (section 6) deserves a thorough re-write to leave the reader with a clear take home message.
